# The relationship between work disability and subsequent suicide or self-harm: A scoping review

**Alex Collie** *, **Shannon Elise Gray**

Healthy Working Lives Research Group, School of Public Health and Preventive Medicine, Monash University, Melbourne, Australia

* alex.collie@monash.edu

## Abstract

Work disability occurs when an injury or illness limits the ability of a worker to participate in employment. While evidence suggests that people with work disability are at increased risk of suicide and intentional self-harm, this relationship has not been the subject of systematic review. This scoping review aims to assess and summarise the research literature regarding the relationship between work disability and subsequent suicide or intentional self-harm. Review protocol was published on the Open Science Foundation and is reported using the Preferred Reporting Items for Systematic Reviews and Meta-Analyses extension for Scoping Reviews. Peer-reviewed studies published in English from 1st January 2000 were included if they reported suicide or self-harm outcomes in people aged 15 years or older with work disability. Studies were identified via systematic search of Medline, Scopus and Pubmed databases, via recommendation from topic experts, and citation searching of included articles. A narrative synthesis was undertaken. Literature search yielded 859 records of which 47 eligible studies were included, nine set in workers' compensation, 20 in sickness absence, 13 in disability pension systems, and five from mixed cohorts. Of 44 quantitative studies, 41 reported a positive relationship between work disability and suicidal behaviour. The relationship is observed consistently across nations, work disability income support systems and health conditions. Several factors elevate risk of suicidal behaviour, including presence of mental health conditions and longer work disability duration. There were few studies in some nations and no suicide prevention interventions. The risk of suicide and self-harm is elevated in people experiencing work disability. Further observational research is required to fill evidence gaps. This review suggests the need for governments, employers and those involved in the care of people with work disability to focus on identification and monitoring of those at greatest risk of suicidal behaviour, and suicide prevention.

## Introduction

Work disability occurs when an injury, illness or other health condition limits the ability of a worker to participate in paid employment. Work disability is a broad concept, encompassing

**Data Availability Statement:** Data extraction tables for the scoping review are available via the publicly accessible Open Science Framework study page - https://osf.io/kh5qf/.

**Funding:** AC received funding from the Australian Research Council via a Future Fellowship (FT190100218). www.arc.gov.au The study funder played no role in the design, analysis, decision to publish or preparation of the manuscript.

**Competing interests:** The authors have declared that no competing interests exist.

partial and temporary loss of work capacity, episodic changes in work capacity, through to total and permanent incapacity [1]. Common diseases and illnesses of working age are the major causes of work disability, and include conditions with high prevalence such as low back pain, depression, anxiety, traumatic injury, as well as cancers and circulatory system disorders [1, 2] The consequences of work disability can be significant for workers. People with prolonged work disability suffer worse mental health [3–5], have shorter life expectancy [6], attend healthcare consultations more frequently with physical symptoms and report higher levels of pain [7], receive more social care [8], and report reduced quality of life [9]. There are also gaps in health service delivery. For example one study of disability pension recipients identified that although 69% reported a diagnosed mental health problem, only 16% reported receiving specialist mental health care [8], a finding reflected in similar studies of workers' compensation cohorts [3, 10].

In addition to treatment for the underlying health condition, people with work disability often require financial support for income losses incurred during periods of time away from work. In nations with developed economies, multiple forms of income support are often available. These can include sickness absence (i.e., sick leave) benefits for temporary work disability offered through employers and sometimes government agencies, and disability benefits for people with permanent or long-term work disability. Some nations also offer financial support through workers' compensation schemes where work is the cause of work disability. Provision of financial support for people with work disability is very common. For example, a recent study in the authors' country of Australia reported that for every 1000 working age Australians, an estimated 49.4 received income support for a period of work disability from a government or private sector (i.e., not their employer) benefit scheme during the 2015/16 year [11].

Suicide and intentional self-harm are major public health problems that affect all ages, especially people of working age. In the author's home nation of Australia, suicide is the leading cause of death in those aged 15 to 44 years, and the third leading cause of premature death [12]. There are more than 33,000 cases of hospitalisation for intentional self-harm annually, with the prevalence higher in younger people and females. People with work disability have multiple risk factors for suicide and suicidal behaviour, some of which are modifiable. First, work disability by definition results in a period of detachment from the workplace, and in some cases may result in unemployment. Unemployment both impairs mental health [13] and is associated with an increased risk of suicide, with the greatest risk within the first five years of unemployment [14]. Second, several studies show that a substantial proportion of people with work disability have mental health conditions, including people in whom the episode of work disability is linked to a physical injury/condition. For example, approximately 30% to 50% of people with workers' compensation claims for musculoskeletal disorders reported moderate to severe psychological distress [3, 4], while mental health conditions are now the most common medical condition among Australians receiving social assistance disability benefits [15]. Third, people with work disability are often involved in administrative benefit systems such as workers' compensation or social assistance schemes. A large proportion of people experience the bureaucratic processes involved in eligibility determination and benefit delivery to be stressful [16, 17]. Qualitative studies demonstrate that in some people, these administrative processes may lead to long-term mental health problems including suicidal ideation, and reduced quality of life [9, 18]. Fourth, work disability is associated with financial distress. People receiving work disability benefits from workers' compensation and social assistance schemes report high levels of financial distress [19]. Financial hardship is frequently cited as a risk factor for suicidal behaviour [20]. Fifth, long periods of work disability are associated with changes in social support networks, including increasing the burden on caregivers and changing the nature of intimate relationships [21]. Social support is a protective factor that reduces the risk of suicide

death in adults [22] that may be adversely impacted during periods of work disability. Finally, many people with work disability and concurrent mental health problems do not receive appropriate mental health treatment, and thus the risk of self-harm or future suicide is more likely to go unrecognised [3, 8].

While the existing evidence is highly suggestive of a link between work disability and later suicide and self-harm, the evidence-base is disaggregated. Studies have been conducted using a variety of methods, in multiple cultural and societal contexts, using a variety of key concept and outcome definitions, and across nations with very differing approaches to supporting people with work disability. There is a need for a scoping study to examine the relationship between work disability and subsequent suicide and self-harm.

The objective of this scoping review is to assess and summarise the research literature regarding the relationship between work disability and subsequent suicide or deliberate self-harm. The review also seeks to answer the question "What personal, psychological, social, medical, environmental or other factors influence the relationship between work disability and subsequent suicide or deliberate self-harm?".

## Materials and methods

This is a scoping review based on the method outlined by Arskey and O'Malley [23] and extended by Levac et al [24]. A protocol for this review was published on the Open Science Foundation website on 29th April 2021 and is available via the following link: https://osf.io/kh5qf/ and provided in S1 Text. We report the scoping review using the Preferred Reporting Items for Systematic Reviews and Meta-analysis extension for Scoping Reviews (PRISMA-ScR). A PRISMA-ScR checklist is provided in S2 Text.

### Eligibility criteria

Peer-reviewed research studies published in English from 1st January 2000 were eligible for inclusion if they reported suicide or self-harm outcomes in people aged 15 years or older who experienced an episode or episodes of work disability. The lower age limit of 15 years is a deviation from the study protocol, in response to identification of studies that included participants from 15 years of age, and that otherwise met all inclusion criteria. Work disability was defined as a complete or partial incapacity to work due to an injury, illness or medical condition that can be temporary or permanent, and resulted in receipt of a financial payment from a third party such as an employer, insurance provider or government agency for a period of absence from work. This could include, for example, a workers' compensation income support benefit, a disability benefit, or a sickness absence payment. Studies reporting samples with injury, illness or medical conditions that affect work capacity but that did not result in a period of work absence were excluded. Suicide and self-harm outcomes were defined as taking or attempting to take one's own life, intentionally harming oneself including suicide attempts or self-injury resulting in hospitalisation, and suicidal thoughts or ideation. Studies that reported on death, injury or harm that arose from the acts of others, by illness or by accidental causes were excluded. No limits on settings or geographic locations were applied.

Consistent with the exploratory nature of the review, we included both qualitative and quantitative primary studies. For quantitative studies, experimental and quasi-experimental study designs including randomized controlled trials, non-randomized controlled trials, before and after studies and interrupted time-series studies were eligible for inclusion. In addition, analytical observational studies including prospective and retrospective cohort studies, case-control studies and analytical cross-sectional studies were considered eligible. Qualitative studies from a variety of theoretical and methodological approaches were eligible including

phenomenological studies, grounded theory, ethnography, qualitative description and action research studies. Case reports, opinion pieces, commentary and literature reviews were excluded from consideration.

## Search strategy

Multiple search methods were used to ensure all relevant prior literature was captured. An initial limited search of PubMed was undertaken to identify articles on the topic. The text words contained in the titles and abstracts of relevant articles, and the index terms used to describe the articles were used to develop a full search strategy, which is provided in S3 Text. The search strategy included a combination of population terms for work disability combined with Boolean OR operators, and outcome terms for suicide and self-harm combined with Boolean OR operators. These two categories were combined with AND operators in final searches.

Search terms were purposefully broad with limited exclusions to ensure all relevant literature was captured. The search strategy, including all identified keywords and index terms, were adapted for each included database or information source, and strategies for each database were developed and refined iteratively. Final searches were conducted in Medline, Scopus and Pubmed on 13th May 2021.

Reference chaining was also conducted, with potentially relevant references identified by a single author (AC) and included in evidence screening. Forward citation searches of included studies were conducted using academic literature databases Scopus and Medline. Backward citation searches from reference lists of included studies were also conducted.

Finally, ten expert academic researchers from North America, Europe and Australia with publications relevant to the topic were contacted and invited to identify studies for inclusion in evidence screening. These topic experts were provided with a summary of the purpose of the review, research questions and the eligibility criteria to guide their recommendations.

An overview of the search and article selection process is provided in Fig 1.

## Selection of sources of evidence

Records identified through database searching, through reference chaining and from expert recommendations were collated and uploaded into EndNote version 9.3 (Clarivate Analytics, PA, USA). Records were compared across data sources and duplicates removed. Article screening proceeded in two major steps. First, titles and abstracts for all records were assessed against the eligibility criteria by two independent reviewers. Records on which both reviewers agreed were excluded or progressed to the next stage. Disagreements between reviewer eligibility ratings, or records on which one or either reviewer could not reach an eligibility decision, were resolved initially through discussion. In cases where agreement could not be reached through discussion, a third reviewer was engaged to make a final decision.

Second, all records that passed the first phase screening were retrieved in full. The full text of documents was assessed by two reviewers independently against the eligibility criteria. Disagreements between reviewer eligibility ratings, or records on which one or either reviewer could not reach an eligibility decision, were resolved through discussion. Reasons for exclusion of sources of evidence at full text were recorded and are reported in the PRISMA-ScR flow diagram presented in Fig 1.

## Data extraction

Data was extracted from eligible records using a data extraction tool developed by the reviewers. The data extracted included specific details about the participants, concept, context, study methods and key findings relevant to the review question/s. Data fields extracted included

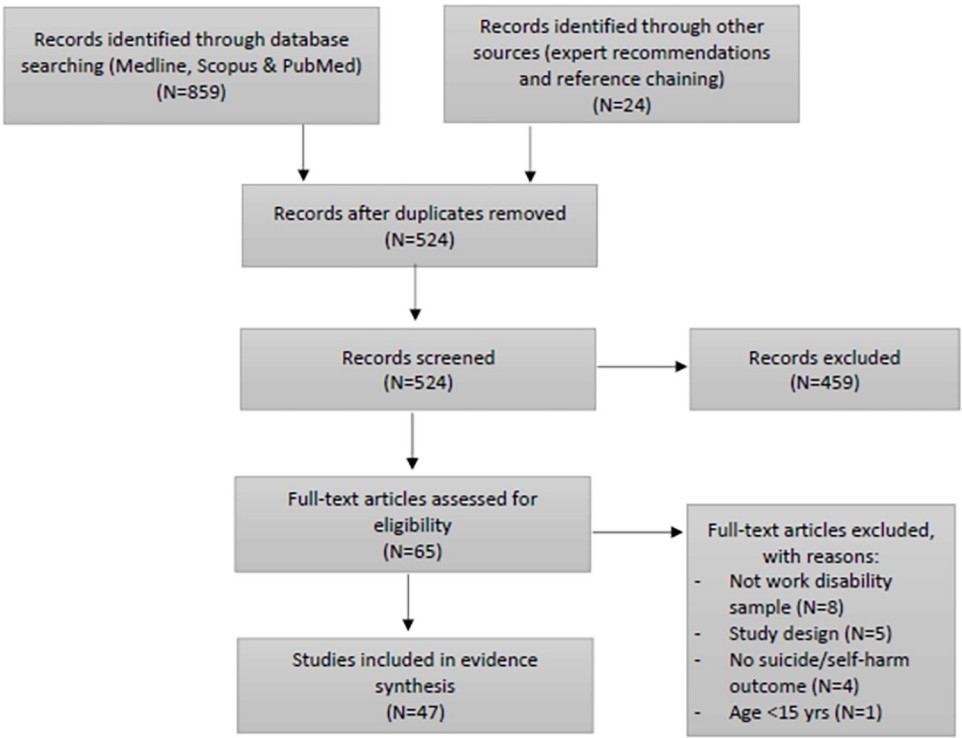

**Fig 1. PRISMA flow diagram.**

study title, authors, journal, year of publication, country of origin, study aim/objective, methodological design, inception period, follow-up period, age distribution, sex/gender distribution, sample size, nature of health condition leading to work disability, type of work disability benefit system (workers' compensation, disability insurance, sickness absence), description of suicide or self-harm outcome/s, statistical analysis method, covariates tested in statistical modelling, whether pre-existing health status was included in statistical adjustment, estimates of statistical effect size, the authors' conclusions regarding the relationship between work disability and outcomes tested, implications for policy and practice, and any gaps identified by study authors. For qualitative studies the major themes relating to suicide and self-harm were extracted and fields related to statistical analysis and effect size were not able to be extracted. Data extraction was initially completed on five eligible studies by a single reviewer (AC). Extracted data was then reviewed by a second reviewer to confirm that all relevant information was extracted. A single reviewer then completed data extraction on the remaining eligible studies. Extracted data is available via the Open Science Foundation project page (https://osf.io/kh5qf/).

## Evidence synthesis

A descriptive numerical summary of included studies was developed and presented in a tabular format for discussion amongst the authors (i.e., an evidence map). The synthesis tables described the numbers of studies by key study features including by study design, country of origin, nature of condition leading to work disability and type of disability benefit system. Inspection of the summary identified three distinct categories of evidence reflecting three unique contexts in which income support was provided to people with work disability, and also reflecting qualitatively different forms of work disability. First, there were a set of eligible

studies set in cause-based workers' compensation (WC) systems, in which the injury or illness leading to work disability must be attributable to employment. Second, there were a set of eligible studies set in disability-based pension (DP) systems in which people typically have long-term or permanent work disability. A third set of eligible studies were set in sickness absence (SA) systems in which people typically have shorter or temporary periods of work disability. There was also a set of studies that included participants receiving either SA or DP benefits and these were combined into a fourth category.

For each of these categories a narrative summary was developed, describing the relationship between work disability and suicide (study main objective), and any factors that contribute to or modify this relationship (key research question). Within each category, studies were grouped and counted according to the study outcome, study design, the health condition leading to work disability, by country of origin and other features. A descriptive summary of findings was developed in groups of studies organised by these features. Gaps in the current evidence base as indicated by lack of included studies or low numbers of studies, were also identified and described in order to identify opportunities for future research.

## Results

The literature database search yielded 859 records, with an additional 24 records identified through reference chaining and expert recommendations. After removing duplicates, a total of 524 unique records proceeded to screening. Application of eligibility criteria in title and abstract screening removed 459 records. Two reviewers disagreed on 58 records (11.1%) of which 25 were resolved through discussion, and the remaining 23 records (4.4%) proceeded to a third reviewer for arbitration. A further 18 were considered ineligible during full-text screening, with two reviewers reaching agreement on all but three records, all of which were resolved via discussion. The remaining 47 records proceeded to data extraction and synthesis.

Of the included studies, 35 reported suicide outcomes, 18 intentional self-harm and a further 8 included suicidal ideation, noting that some studies reported multiple outcomes. A single study reported a composite outcome that incorporated suicide with intentional self-harm. Thirty-two of the eligible studies reported an objective of assessing the relationship between work disability and later suicide or self-harm. In the remaining 15 studies this relationship was reported in statistical analysis but was not an explicit objective of the study.

Methods were heterogenous, with 31 studies reporting either prospective or retrospective cohort designs, 6 case-control studies, 5 cross-sectional studies, 3 qualitative studies and a further 2 ecological studies. Thirty-five of the studies involved linkage of data between two separate data sources, and thirty-seven included data from more than one source. The most common data sources were administrative records (N = 37, e.g., social security payment records, workers' compensation records) and electronic health records (N = 32, e.g., hospital records), while 9 reported data collected via participant or healthcare provider surveys, and 3 from participant interviews. Among the 44 quantitative studies, the primary statistical analysis methods included survival analysis (N = 22), logistic regression (N = 11), Poisson regression (N = 2), linear regression (N = 2) and calculation of standardised mortality ratio (N = 3). A further 7 studies reported other primary analysis methods. The three qualitative studies reported results of thematic analysis.

The majority of included studies (N = 28) did not focus on populations with specific health conditions. Eleven studies were conducted in samples with mental health conditions, a further four in samples with musculoskeletal disorders and four studies in samples with specific medical conditions. In 21 studies an indicator of health status prior to the onset of work disability was included as a confounder in the final statistical model. Forty of the eligible studies were in

community-based samples, four in groups defined on the basis of their occupation, and three in groups defined on the basis of their medical/hospital admission status. Three studies were focused on young adults, one on middle aged adults and the remainder included samples across the working age spectrum.

## Studies in workers' compensation benefit recipients

Nine eligible studies reported relevant outcomes in people with work disability who had received financial support from workers' compensation schemes. The studies are summarised in Table 1. Five of the studies reported suicide outcomes, four suicidal ideation and a single reported intentional self-harm. Eight of the studies included samples of people with musculoskeletal disorders, with three focused solely on this condition and five also including people with accepted workers' compensation claims for other conditions. A single study reported on people with traumatic injury and specifically excluded those with gradual onset musculoskeletal disorders or intentional self-harm as the cause of work disability.

Two studies from the USA state-based workers' compensation schemes report greater risk of suicide among injured workers with compensation claims involving time off work compared with workers whose claims were for medical expenses only [25, 26]. Applebaum et al [25] reported a retrospective cohort study of 100,806 adults with illness and injury acquired in the course of work between 1994 and 2000. Over a median 16 years follow-up period, those with at least 7 calendar days of lost time following their injury demonstrated a 92% and 72% increase in mortality hazard due to suicide for women and men, respectively. In a similar study, Martin et al [26] reported an 85% increase in suicide mortality hazard among workers with low back pain during a study with an 18-year follow-up period, among workers injured in 1998 or 1999. One Korean and one Taiwanese study reported an approximate doubling of suicide rates among adults with workers' compensation claims following musculoskeletal conditions [27] or traumatic injury [28] compared with national suicide mortality reference data. A single cross-sectional study of rehabilitation patients with chronic or acute pain conditions receiving workers' compensation reported significantly higher relative risk of suicidal ideation compared to similar patients not receiving workers' compensation [29]. For example, the relative risk (RR) of "wanting to die because of pain" was 4.22 (2.67,6.68). An Australian death register study identified that 29% of workers who had died by suicide in the state of Victoria were involved in a workers' compensation claim [30].

The literature search identified three qualitative studies in which workers reported suicidal thoughts [18, 31, 32]. These were all conducted in Canadian workers' compensation settings. The author of one study reports a surprisingly high frequency of discussions of suicide among included workers, and also that other participants in workers compensation such as lawyers also report suicide risk among their workers clients. Workers involved in these studies link suicidal thoughts to their experiences of prolonged and serious disability, precarious personal and financial situations, and also to procedures, policies and attitudes of compensation agencies such as conducting surveillance and complex and dysfunctional administrative processes.

Some notable gaps in this literature include a small number of studies with longitudinal designs, undertaken in a small number of jurisdictions; and the lack of studies on workers with mental health condition claims.

## Studies in sickness absence benefit recipients

Twenty eligible studies reported relevant outcomes in people with work disability who had received sickness absence benefits. The studies are summarised in Table 2. Fifteen studies reported suicide outcomes, seven intentional self-harm, two suicidal ideation and a single

**Table 1. Summary of workers' compensation studies.**

| Author (Year) & Location | Study Design & Follow-up period | Data Sources | Sample size & percent female | Sample definition (exposed group) | Sample definition (comparator group) | Primary outcome/s | Main finding |
|---|---|---|---|---|---|---|---|
| Applebaum et al (2019) New Mexico, USA | Prospective Cohort Study 13 to 18 years | Workers' compensation; Social security; Coronial data | N = 100,806 Female 38% | Individuals (15 to 80 years) with injuries resulting in more than 7 days off work ("lost time injuries") | Individuals with injuries resulting in 7 or fewer days off work who received only medical benefits ("medical only" injuries) | Suicide Defined using ICD-10 coding from national death index | "Lost-time injuries were associated with mortality related to suicide. Among women, lost time injuries were associated with a 92% increase in mortality hazard due to suicide. For men, lost time injury was associated with a 72% increased hazard of suicide." |
| Fishbain et al (2009) United States of America | Cross sectional study Not applicable | Participant questionnaire | N = 2,264 Sex distribution not reported | Adult rehabilitation patients (18 to 65 years) with acute or chronic pain | Adult rehabilitation patients (18 to 65 years) with acute or chronic pain but no compensation claim | Suicidal ideation 1. History of wanting to die 2. Wanting to die because of pain 3. Recent frequent suicidal ideation 4. Has suicide plan  Intentional self-harm 5. History of suicide attempt | "Being a rehabilitation patient and having worker's compensation status increased the risk over just simply being a rehabilitation patient for all suicidality items, except "history of wanting to die."" |
| Lee et al (2021) Korea | Prospective Cohort Study 1 to 13 years | Workers compensation; Death register | N = 775,537 Female 20.2% | Individuals aged 15 to 79 years with work-related injury, excluding those with MSD or intentional self-harm | National suicide mortality reference data | Suicide Defined using ICD-10 coding from national death register | "Overall, occupationally injured workers showed significantly higher mortality compared with the reference population.  We found that occupationally injured workers have a higher risk of suicide both in men and women. Contrary to our hypothesis, injured workers with disabilities had lower rates of suicide compared with non-disabled workers." |
| Lin et al (2010) Taiwan | Retrospective Cohort Study 1 to 20 years | Death register; Workers' compensation | N = 71,001 Sex distribution not reported | Adults (age not stated) with upper or lower limb injury | National suicide mortality reference data | Suicide Defined using ICD-10 coding from national death register | "In this study, lower extremity amputees were found to have about a 4-fold greater risk of intentional self-harm [resulting in death], and it corroborated previous reports suggesting that people with severe lower extremity injuries and/or amputations usually suffer considerable psychological distress often leading to alcohol addiction and substance abuse." |

*(Continued)*

**Table 1.** (Continued)

| Author (Year) & Location | Study Design & Follow-up period | Data Sources | Sample size & percent female | Sample definition (exposed group) | Sample definition (comparator group) | Primary outcome/s | Main finding |
|---|---|---|---|---|---|---|---|
| Lippel (2007) Quebec, Canada | Qualitative study Not applicable | Participant interviews | N=85 Female 48% | Adults with workers' compensation claims | Not applicable | Suicidal ideation Self-report via interview | "The fact that design and implementation practices of social security systems can affect the health of those they are meant to help is well known ...yet these issues are rarely discussed with regard to disability insurance systems and workers' compensation. This study confirms that they are just as relevant in this context and require attention of both researchers and policy makers." |
| MacEachen et al (2007) Canada | Qualitative study Not applicable | Participant interviews | N=37 Female 62.2% | Adults aged 30 to 69 years with conditions including MSD, respiratory, brain injury and cancer. | Not applicable | Suicidal ideation Self-report via interview | "Their emotional situation [of disabled workers] was often precarious. Workers with injuries who attended the peer support groups usually had personal and financial situations that had reached a critical point, leaving them feeling trapped in lives over which they had little control." "Many of these workers were angry, depressed, or alienated from their former sources of support including their families." "Several participants gave detailed accounts of how it would be easier to put an end to their dire financial and personal situations by committing suicide." |
| MacEachen et al (2010) Ontario, Canada | Qualitative study Not applicable | Participant interviews | N=69 Female 45.8% | Adults (age not specified) with conditions including MSD, respiratory, brain injury and cancer. | Not applicable | Suicidal ideation Self-report via interview | "This study links workers' and providers' accounts of problematic [return to work] experiences to policies and procedures that seemed unable to facilitate the recognition and management of these issues. It provides an explanation for the poor health of workers with long-term workers' compensation claims in a confluence of seemingly mundane, but dysfunctional RTW processes, that can slow the course of a claim and accumulate to create a damaging 'toxic dose' to some workers." |

(*Continued*)

**Table 1.** (Continued)

| Author (Year) & Location | Study Design & Follow-up period | Data Sources | Sample size & percent female | Sample definition (exposed group) | Sample definition (comparator group) | Primary outcome/s | Main finding |
|---|---|---|---|---|---|---|---|
| Martin et al (2020) West Virginia, USA | Prospective Cohort Study 16 to 18 years | Workers' compensation; Social security; Death register | N=14,218 Female 37.1% | Adults (age not specified) with low back pain receiving time-loss benefits | Adults with low back pain receiving medical benefits only | Suicide Defined using ICD-10 coding from national death register | "Mortality from intentional self-harm were significantly associated with lost time [injuries] only. Hazard Ratios for intentional self-harm according to amount of lost time, permanent disability, percent permanent disability, and surgical treatment were also elevated, but were not statistically significant." |
| Routley et al (2012) Victoria, Australia | Retrospective cohort study Not reported | Coronial records | N=62 Female 16.2% | Adults aged 25 years and over completing suicide (injury and illness leading to work disability not specified) | None | Suicide Defined using coronial records | "At least 22 of the suicides were in receipt of benefits from, or under investigation by, compensation agencies at the time of their suicide, and this was frequently implicated within case reports as being a significant source of stress for them. Stress attributable to involvement in legal processes is also indicated in the literature as being a contributory factor to suicide in both acute and chronic post-injury phases." |

Note: N = Number; MSD = Musculoskeletal disorders. ICD-10 = International Classification of Diseases version 10.

reported a composite outcome including suicide, self-harm and ideation. Thirteen studies did not specify a specific condition and included samples of people with sickness absence from diverse causes, six studies included samples of people with Mental Health Conditions and one a musculoskeletal disorder sample.

Overall, included studies demonstrate an elevated risk of suicide and self-harm among people with a sickness absence spell. Nineteen of the 20 eligible studies observed positive associations between sickness absence and later suicide or self-harm.

Eight prospective or retrospective cohort studies showed a higher rate of suicide during follow-up among people with a sickness absence spell during the study inception period, compared to those without sickness absence [33–40]. Multiple of these studies observed that people with longer sickness absence spells, and those with multiple spells, are at increased risk of suicide than those with short or no absences [33, 34, 37, 39], and that those whose absence is due to mental health conditions are at increased risk of suicide [35–39]. Two of these cohort studies observed similar associations between mental health related sickness absence, and sick leave duration, and intentional self-harm [39, 40], while other studies also observed that there was increased rate of suicide or intentional self-harm in people with sickness absence due to musculoskeletal disorders [36, 39] or other conditions including injury/poisoning, digestive, circulatory, nervous and respiratory disease [39].

Three retrospective case-control studies observed that a prior sickness absence spell was a significant risk factor for hospitalisation for self-harm or suicide [41–43], although this effect

**Table 2. Summary of sickness absence studies.**

| Author (Year) & Location | Study Design & Follow-up period | Data sources | Sample size & Percent female | Study (Exposure) Group | Comparison Group | Primary outcome/s | Main finding |
|---|---|---|---|---|---|---|---|
| Ando et al (2013) Tokyo, Japan | Cross sectional study Not applicable | Participant questionnaire; Health practitioner questionnaire | N = 189 Female 45.2% | Adults (>19 years) receiving outpatient treatment for depression | None | Suicidal ideation As reported by treating psychiatrists | This study showed that taking sick-leave was associated with **decreased odds** of current suicidal ideation in people receiving outpatient treatment for depression. |
| Billingsley et al (2020) Sweden | Prospective Cohort Study 15 years | Social security; Death register; Hospital records; Social / labour market statistics | N = 27,178,226 person years Female 58.1% | Adults (<66 years) who received 1 or more days of paid sickness absence benefit payments in 1996. | Adults (<66 years) who did not receive any sickness absence benefit payments in 1996. | Suicide Based on ICD-10 codes in Death Register | This study shows a gradient in the relationship between sickness absence and suicide mortality. In both men and women, longer durations of sickness absence were associated with a higher hazard ratio for suicide. |
| Bjorkenstam et al (2014) Sweden | Prospective Cohort Study 11 years | Social security; Death register; Hospital records; Social / labour market statistics | N = 4,669,235 Female 48.7% | Adults (20 to 64 years) who received 1 or more days of sickness absence benefit payments in 1995. | Adults (20 to 64 years) who did not receive any sickness absence benefit payments in 1995. | Suicide "Cause-specific mortality" derived from Death Register | This study shows that sick leave was associated with a higher risk of suicide in both women and men. The relative risk for suicide increased in a graded fashion with more sick-leave days. Adjusting for prior inpatient care decreased the estimates significantly. |
| Braquehais et al (2020) Catalonia, Spain | Prospective Cohort Study Not reported | Electronic medical records | N = 1214 Female 70.7% | Physicians and nurses (age range not stated) with mental health conditions enrolled in a treatment program | None | Suicidal ideation High risk of suicide identified using the Mini International Neuropsychiatric Interview | This study observed that physicians and nurses with a high risk of suicide were more frequently on sick leave when admitted to the treatment programme. |
| Bryngelson et al (2013) Sweden | Prospective Cohort Study 8 years | Social security; Death register; Hospital records; Social / labour market statistics | N Cohort 1 (1990) = 244,990 Female 70.7% N Cohort 2 (2000) = 764,137 Female 77.1% | Individuals (16 to 60 years) with psychiatric disorders and at least one long spell of sickness absence exceeding 90 days, beginning in 1990 (Cohort 1) or 2000 (Cohort 2). | Employed individuals (16 to 60 years) with no registered sick leave in 1990 (Cohort 1) or 2000 (Cohort 2) | Suicide Based on ICD-10 codes in Death Register | Employees with long-term sickness absence due to a psychiatric disorder were at increased risk of suicide mortality compared to the reference group. This elevated risk remained after controlling for prior inpatient care. |
| Dorner et al (2017) Sweden | Prospective Cohort Study 4 years | Social security; Death register; Hospital records; Social / labour market statistics | N = 66,097 Female 69.2% | Adults (18 to 59 years) with new sickness absence spells due to common mental health conditions during 2006. | None | Intentional self-harm Based on ICD-10 codes from hospital records | Among people with sickness absence due to mental health conditions, a history of specialised outpatient care due to mental disorders, inpatient care regardless of diagnosis, and having been prescribed medication were associated with a higher risk of future suicide attempt. |

*(Continued)*

**Table 2.** (Continued)

| Author (Year) & Location | Study Design & Follow-up period | Data sources | Sample size & Percent female | Study (Exposure) Group | Comparison Group | Primary outcome/s | Main finding |
|---|---|---|---|---|---|---|---|
| Ishtiak-Ahmed et al (2013) Sweden | Prospective Cohort Study 3 to 4 years | Social security; Death register; Hospital records; Social / labour market statistics | N = 36,304 Female 72.5% | Individuals (16 to 64 years) with at least one sickness absence spell due to a stress-related mental health condition in 2005 | None | Suicide Based on ICD-10 codes in Death Register Intentional self-harm Based on ICD-10 codes from Hospital Records | Young age, low education, being a single parent, higher numbers of days and spells of sickness absence, and inpatient and outpatient care were predictive of suicide attempt. Previous and on-going inpatient and outpatient care due to mental diagnoses and having two compared with only one sickness absence spell predicted suicide mortality. |
| Jansson et al (2012) Sweden | Prospective Cohort Study 3 years | Social security; Death register; Hospital records; Social / labour market statistics | N = 4,760,987 Female 48.2% | Adults (20 to 64 years) who received payment for at least 1 sickness absence day due to a musculoskeletal diagnoses in 2005. | Adults (20 to 64 years) who did not receive any sickness absence benefits in 2005. | Suicide Based on ICD-10 codes in Death Register | Increased risks of suicide among people with sickness absence due to musculoskeletal disorders. The increased risk was higher for people with dorsopathies than for people with arthropathies and connective tissue diseases, or for people with soft tissue disorders, osteopathies and chondropathies. This study also observed stronger associations between sickness absence due to non-musculoskeletal conditions and suicide mortality. |
| Leira et al (2020) North-Trondela County, Norway | Prospective Case Control Study 19 years | Death register; Participant survey; Hospital records | N = 66,140 Female 53.8% | Adults (20+ years) with a record of hospitalisation for self-harm or completing suicide between 1995 and 2014. | Adults (20+ years) without a record of hospitalisation for self-harm or completing suicide between 1995 and 2014. | Suicide Based on ICD-10 codes in Death Register Intentional self-harm Based on ICD-10 codes from Hospital Records | This study observed that sickness absence during the previous 12 months was almost twice as high in people who were hospitalised following intentional self-harm than in people who had completed suicide and the group without a record of hospitalisation or suicide. |

(*Continued*)

**Table 2.** (Continued)

| Author (Year) & Location | Study Design & Follow-up period | Data sources | Sample size & Percent female | Study (Exposure) Group | Comparison Group | Primary outcome/s | Main finding |
|---|---|---|---|---|---|---|---|
| Lunde et al (2021) Norway | Retrospective Case control Study not applicable | Death register; Hospital records; Social security | N = 9,853 cases Female 45.6% N = 186,092 controls Female % not reported | Young adults (18 to 35 years) hospitalised for deliberate self harm between 2008 and 2013. | Community controls matched on age and gender and without a record of hospitalisation for deliberate self-harm. | Intentional self-harm Based on ICD-10 codes from hospital records | A prior spell of sickness absence was observed to be a statistically significant risk factor for later hospitalisation due to intentional self-harm, if the sickness absence was due to a psychiatric disorder. The risk of hospitalisation for intentional self-harm was greater for people currently sickness absent, regardless of cause, but was greater if the sickness absence was due to a psychiatric disorder. There was a small additional increase in risk for those who had had 3 or more sick leave spells. |
| Lundin et al (2012) Sweden | Prospective Cohort Study 2 to 4 years | Death register; Social security; Census data | N = 771,068 Female 49.8% | Adult (aged 25 to 58 years) residents of Stockholm County in 1990-1991 | None | Suicide Based on ICD-10 codes in Death Register | For both men and women, longer durations of sickness absence were associated with statistically higher odds of suicide. This effect was stronger in men than in women, and for people with sickness absence exceeding 62 days. Adjusting for unemployment had very little effect on the association between sickness absence and suicide. Thus, unemployment was not on the causal pathway (had no mediating effect) in either men or women. |
| Melchior et al (2010) France | Prospective Cohort Study 16 years | Death register; Employer records | N = 19,962 Female 26.8% | Employees (35 to 50 years) of a gas and electricity company with medically certified sick-leave spells exceeding 7 days between 1990 and 1992 | Employees (35 to 50 years) of a gas and electricity company without sick-leave spells between 1990 and 1992 | Suicide Based on ICD-10 codes in Death Register | This study observed that workers with sickness absence due to a psychiatric disorder were at increased risk of death from suicide. Study participants who were sickness absent for at least 7 days due to a psychiatric disorder had a 6-fold excess risk of suicide. |

(*Continued*)

**Table 2.** (Continued)

| Author (Year) & Location | Study Design & Follow-up period | Data sources | Sample size & Percent female | Study (Exposure) Group | Comparison Group | Primary outcome/s | Main finding |
|---|---|---|---|---|---|---|---|
| Mittendorfer-Rutz et al (2012) Sweden | Prospective cohort study 3 to 4 years | Social security; Death register; Hospital records | N = 4,857,943 Female 47.5% | People (16 to 64 years) with at least one new sick-leave spell during 2005 | People (16 to 64 years) without a sick-leave spell during 2005 | Suicide Based on ICD-10 codes in Death Register | The risk of suicide was increased nine-fold in people with a new sick-leave spell due to a mental health condition, after adjusting for age and sex. This risk estimate remained three-fold increased after controlling for a broader range of socio-demographic factors and prior health service use. |
| Roskar et al (2020) Slovenia | Ecological study Not applicable | Death register; Social / labour market statistics; Participant questionnaire; Social security | 212 municipalities | Population of Slovenia The exposure variable was the number of sick leave days per capita (at area level) in general and due to a mental health condition. | Not applicable | Suicide Based on ICD-10 codes in Death Register | This study observed that the number of sickness absence days per capita was statistically associated to the rate of suicide in local areas in a univariate analysis, but not in multivariate analysis adjusting for socio-economic, behavioural and health service characteristics. |
| Szlejf et al (2020) Brazil | Cross sectional study Not applicable | Employer records | N = 16,890 Female 69.8% | Employees (18 + years) of a health organisation with sick leave spells during 2017 or 2018. | Employees (18 + years) of a health organisation with no sick leave spells during 2017 or 2018. | Composite outcome Suicide, suicidal ideation or self-harm based on employee health records | Employees with sickness absence from any cause had around 7 times higher odds of the composite outcome than employees without any sickness absence spell, after adjusting for age, sex, education and job position. This effect was strongest among employees with sickness absence due to mental health conditions, and was also elevated among employees with more than 14 days of sickness absence. |

(*Continued*)

**Table 2.** (Continued)

| Author (Year) & Location | Study Design & Follow-up period | Data sources | Sample size & Percent female | Study (Exposure) Group | Comparison Group | Primary outcome/s | Main finding |
|---|---|---|---|---|---|---|---|
| Tang et al (2019) Norway | Retrospective Case Control Study Not applicable | Death register; Social security; Social / labour market statistics | N = 9,313 Cases Female 27.3% N = 169,235 Controls Female % not reported | Adults (18 to 66 years) completing suicide between 1992 and 2012 | Community controls matched on aged and gender | Suicide Based on ICD-10 codes in Death Register | Among males, a history of physical illness resulting in sickness absence was associated with significantly greater odds of subsequent suicide. The odds of suicide increased with a greater number of sickness absence spells, and longer durations of sickness absence. Effects remained significant when statistical models were adjusted for history of mental illness and socioeconomic status. Among females the opposite was observed. A history of sickness absence due to physical illness, multiple sickness absence spells and longer duration of sickness absence were all associated with a significantly reduced odds of subsequent suicide. |
| Vahtera et al (2004) Finland | Prospective Cohort Study 1 to 6 years | Death register; Employer records | N = 41,736 Female 69.3% | Municipal employees (18 + years) with a job contract lasting at least five consecutive years between 1990 and 2000. | None | Suicide Based on ICD-10 codes in Death Register | This study found that employees with more than one long sickness absence spell (exceeding 3 days) per annum had more than 7 times higher mortality from suicide. Shorter term absences of 1 to 3 days were not significantly associated with suicide. Employees with more than 15 days sickness absence per annum had more than 3 times greater odds of suicide. |

**Table 2.** (Continued)

| Author (Year) & Location | Study Design & Follow-up period | Data sources | Sample size & Percent female | Study (Exposure) Group | Comparison Group | Primary outcome/s | Main finding |
|---|---|---|---|---|---|---|---|
| Wang et al (2014) Sweden | Prospective Cohort Study 6 years | Social security; Death register; Hospital records; Social / labour market statistics | N = 4,923,404 Female 47.5% | Individuals (16 to 64 years) with at least one new sick-leave spell during 2005 | Individuals (16 to 64 years) without a new sick-leave spell during 2005 | Suicide Based on ICD-10 codes in Death Register Intentional self-harm Based on ICD-10 codes from Hospital Records | Women and men with sickness absence showed a twofold increase in hazard ratio of both intentional self-harm and suicide compared with people without any sickness absence. The association increased with increasing sickness absence duration. Sickness absence resulting from mental health conditions was most strongly associated with both suicide and intentional self-harm among both women and men. Other specific sickness absence diagnoses associated with intentional self-harm included injury/ poisoning, musculoskeletal, digestive, circulatory, nervous and respiratory diseases. For suicide, individuals with sickness absence due to musculoskeletal disorders and injury/ poisoning had increased risk. |
| Wang et al (2015) Sweden | Prospective Cohort Study 6 years | Social security; Death register; Hospital records; Social / labour market statistics | N = 21,096 Female 59.9% | Individuals (16 to 64 years) with depression who received psychiatric in or outpatient care during 2005 | None | Suicide Based on ICD-10 codes in Death Register Intentional self-harm Based on ICD-10 codes from Hospital Records | Among patients with depressive disorders, significantly elevated risk of intentional self-harm was associated with new sickness absence spells, full time sickness absence, long sick-leave length, one or more sickness absence spells, as well as sickness absence due to mental health conditions. |

*(Continued)*

**Table 2.** (Continued)

| Author (Year) & Location | Study Design & Follow-up period | Data sources | Sample size & Percent female | Study (Exposure) Group | Comparison Group | Primary outcome/s | Main finding |
|---|---|---|---|---|---|---|---|
| Wang et al (2016) Sweden | Prospective Cohort Study 2 years | Social security; Death register; Hospital records; Social / labour market statistics | N Cohort 1 (2006) = 4,071,935 Female 46.1% N Cohort 2 (2009) = 4,176,873 Female 46.8% | Individuals (16 to 64 years) with at least one new sick-leave spell during 2006 (Cohort 1) or 2009 (Cohort 2) | Individuals (16 to 64 years) with no new sick-leave spell during 2006 (Cohort 1) or 2009 (Cohort 2) | Suicide Based on ICD-10 codes in Death Register Intentional self-harm Based on ICD-10 codes from Hospital Records | This study observed that sickness absence due to any cause, and absence due to common mental health conditions were associated with both suicide and intentional self-harm. In addition, longer sickness absence duration was associated with intentional self-harm. |

Note: N = Number; MSD = Musculoskeletal disorders. ICD-10 = International Classification of Diseases version 10.

was only significant among males in one of these studies [43]. Two studies identified predictors of suicide or self-harm among cohorts of people with sickness absence as including absence due to mental health conditions, a history of healthcare due to mental illness, young age, low education, being a single parent and having a high number of sickness absence days [44, 45]. One study identified predictors of suicide among adult residents of Stockholm as including longer duration of sickness absence, with a stronger effect in men than women [46]. Another study of people receiving healthcare for depression during 2005 identified elevated risk of later self-harm resulting in hospitalisation among people with new, full-time or long sickness absence spells [47]. Three studies were conducted in specific occupational cohorts. These identified that physicians and nurses with a high risk of suicide were more frequently on sick leave when admitted to an inpatient treatment program [48]; that government employees with more than one sickness absence spell exceeding 3 days had more than 7 times higher mortality from suicide, while employees with more than 15 days sickness absence per annum had 3 times the odds of suicide [49]; and that employees of a healthcare organisation with sick leave during 2017 or 2018 had seven times the odds of recording a composite outcome encompassing suicide, self-harm or suicidal ideation, than employees with no sick leave spells [50].

A single population based ecological study in Slovenia observed a univariate relationship between sickness absence duration per capita and the rate of suicide at a local area level, however this effect was not observed in statistical models that adjusted for area level socioeconomic and health service characteristics [51]. Finally, a single cross-sectional study reported that taking sick leave was associated with decreased odds of current suicidal ideation in people receiving outpatient treatment for depression [52].

Some notable gaps in this literature include relatively few (N = 5) studies outside Nordic countries and no studies in North America or Asia; few studies in specific occupational settings or groups, relatively fewer studies examining outcomes of suicidal ideation or self-harm; and a lack of qualitative studies.

## Studies in disability pension recipients

Thirteen eligible studies reported relevant outcomes in people with work disability who had received a disability pension. The studies are summarised in Table 3. Eleven studies reported suicide outcomes, seven intentional self-harm and two suicidal ideation. Studies were from six

**Table 3. Summary of disability pension studies.**

| Author (Year) & Location | Study Design & Follow-up period | Data sources | Sample size & percent female | Sample definition (exposed group) | Sample definition (Comparison group) | Primary outcome/s | Main finding |
|---|---|---|---|---|---|---|---|
| Barr et al (2016) England | Ecological study Not applicable | Participant questionnaire; Social security; Labour force statistics; Death register | 1.03 million Female % not stated | Adults (18 to 64 years) receiving disability pension whose eligibility was reassessed via a work capacity assessment. | Not applicable | Suicide Area level, annual age-adjusted mortality rates from suicide and injury of undetermined cause | This area-level study found that local areas where a greater proportion of the population were exposed to DP work capacity reassessment process experienced a greater increase in suicides. These associations were independent of baseline conditions in these areas, and the increases followed rather than preceded, the reassessment process. |
| Becker et al (2009) Florida, USA | Retrospective Case Control Study Not applicable | Medicaid eligibility and claim files | N=1,305 Female 63.6% | Adults (age range not stated) enrolled in medicare and also receiving supplementary security income | Adults (age range not stated) enrolled in medicare but not receiving supplementary security income | Suicide Identified from Medicaid files | This study found significantly greater odds of suicide in the exposed group receiving SSI than in the comparison group.<br><br>The three strongest predictors of suicide in the exposed group were psychiatric hospitalization, having an involuntary psychiatric examination, or receiving extensive behavioural health services. |
| Bjorkenstam et al (2014) Sweden | Prospective Cohort Study 14 years | Social security; Death register; Hospital records; Social / labour market statistics | N=5,006,523 Female 48.8% | Individuals (16 to 64 years) granted full or part-time disability pension in 2004. | Individuals (16 to 64 years) not in receipt of disability pension in 2004. | Suicide Based on ICD-10 codes in National Death Index | This study found a significantly higher hazard of suicide among both male and female DP recipients compared to control group.<br><br>The risk of suicide was highest in DP recipients with mental health diagnoses (men and women) and also significantly higher than control in those with musculoskeletal (women) and neurological (men) diagnoses. |
| Bjorkenstam et al (2015) Sweden | Prospective Cohort Study 6 years | Social security; Death register; Hospital records; Social / labour market statistics | N=11,346 Female = 56.4% | Individuals (16 to 64 years) with Multiple Sclerosis receiving disability pension in 2004 | Individuals (16 to 64 years) with Multiple Sclerosis not receiving disability pension in 2004 | Suicide Based on ICD-10 codes in Death Register Suicide attempts Based on ICD-10 codes in Hospital Records | This study reported a non-significant tendency towards a higher risk for suicidal behaviour in people with multiple sclerosis receiving the DP. |

*(Continued)*

**Table 3.** (Continued)

| Author (Year) & Location | Study Design & Follow-up period | Data sources | Sample size & percent female | Sample definition (exposed group) | Sample definition (Comparison group) | Primary outcome/s | Main finding |
|---|---|---|---|---|---|---|---|
| Butterworth et al (2006) Australia | Cross-sectional study Not applicable | Participant questionnaire | N=8,533 Female = 48.4% | Adults (18 to 64 years) not in the labour force receiving the Disability Support Pension as their main source of income. | Adults not reliant on welfare payments | Suicidal ideation self-reported suicidal ideation in past 12 months<br><br>Intentional self-harm self-reported attempted suicide in past 12 months | This study reported that people not in the labour force (most of whom receive DP benefits) were more likely to report suicidal behaviours than the comparison group. |
| Ehn et al (2016) Sweden | Cross-sectional study Not applicable | Medical records; Participant questionnaire | N=67 Female = 53% | Adults (18 to 65 years) with Usher syndrome in receipt of disability pension | Adults (18 to 65 years) with Usher syndrome not receiving the disability pension | Suicidal ideation self-reported suicidal thoughts<br><br>Intentional self-harm self-reported attempted suicide | This study reported that participants with Usher syndrome receiving the DP were significantly more likely to report having made at least one suicide attempt, and having a history of suicidal thoughts. |
| Jonsson et al (2013) Sweden | Prospective Cohort Study 5 years | Social security; Death register; Hospital records; Social / labour market statistics | N=525,276 Female = 48.7% | Young adults (19 to 23 years) receiving a disability pension in 2005 | Young adults (19 to 23 years) NOT receiving a disability pension in 2005 | Intentional self-harm Based on ICD-10 codes in Hospital records<br><br>Suicide Based on ICD-10 codes from Death register | Young adults receiving DP with a mental health condition have a statistically higher risk of both suicide attempts and completed suicide. Particularly high hazard ratios were observed for those with bipolar affective disorder and personality disorders.<br><br>Young adults receiving DP with a somatic condition have a statistically higher risk of suicide attempt, but not completed suicide. |
| Jonsson et al (2014) Sweden | Prospective Cohort Study 5 years | Social security; Death register; Hospital records; Social / labour market statistics | N Cohort 1 (1995) = 559,147 Female 48.9% N Cohort 2 (2000) = 504,741 Female 48.9% N Cohort 3 (2005) = 525,276 Female 48.7% | Young adults (19 to 23 years) receiving a disability pension benefit in 1995, 2000 or 2005 | Young adults (19 to 23 years) NOT receiving a disability pension benefit in 1995, 2000 or 2005 | Intentional self-harm Based on ICD-10 codes in Hospital records<br><br>Suicide Based on ICD-10 codes from Death register | Young adults receiving DP due to psychiatric diagnoses had statistically elevated risk of suicide and suicide attempt relative to comparison group in three separate cohorts.<br><br>Young adults receiving DP due to somatic diagnoses had statistically elevated risk of suicide attempt relative to comparison group in three separate cohorts, but not suicide. |
| Kiviniemi et al (2011) Finland | Prospective Cohort Study 5 years | Social security; Death register; Hospital records | N=3,875 Female = 41.3% | Individuals (16 to 65 years) with hospital admission due to schizophrenia receiving a Disability Pension during the 5-year period after the onset of schizophrenia. | Individuals (16 to 65 years) with hospital admission due to schizophrenia NOT receiving a Disability Pension during the 5-year period after the onset of schizophrenia. | Suicide Based on ICD-10 codes from Death register | This study found a significantly decreased likelihood for suicide mortality among people with schizophrenia receiving DP compared with schizophrenia without DP. |

*(Continued)*

**Table 3.** (Continued)

| Author (Year) & Location | Study Design & Follow-up period | Data sources | Sample size & percent female | Sample definition (exposed group) | Sample definition (Comparison group) | Primary outcome/s | Main finding |
|---|---|---|---|---|---|---|---|
| Leinonen et al (2014) Finland | Prospective Cohort Study 11 years | Social security; Death register | N=392,985 Female = 45.1% | Adults (25 to 64 years) not work disabled in 2006 who receive a new disability pension due to common mental health conditions between 1997 and 2007. | Other adults (25 to 64 years) not work disabled in 2006. | Suicide Based on ICD-10 codes from Death register | This study shows significantly higher mortality from suicide among DP recipients due to depression and other mental health conditions than among the population in general, among both male and female DP recipients. Hazard ratios for suicide were higher in female than in male DP recipients. |
| Rahman et al (2014) Sweden | Prospective Cohort Study 5 years | Social security; Death register; Hospital records; Social / labour market statistics | N=46,745 Female = 66.3% | Adults (18 to 64 years) with common mental health conditions receiving a disability pension in 2005 | none | Intentional self-harm Based on ICD-10 codes in Hospital records  Suicide Based on ICD-10 codes from Death register | Among people receiving DP due to mental health conditions, a higher risk of suicide attempt was observed among females, those with a prior suicide attempt, low education, living alone, younger age, prior inpatient care, prescribed antidepressants and anxiolytics.  A higher risk of suicide was observed among males, those with prior suicide attempt, living alone, prescribed antidepressants and anxiolytics. |
| Rahman et al (2016) Sweden | Prospective Cohort Study 5 years | Social security; Death register; Hospital records; Social / labour market statistics | N=46,515 Female = 66.4% | Adults (18 to 64 years) with common mental health conditions receiving a disability pension in 2005 | none | Intentional self-harm Based on ICD-10 codes in Hospital records  Suicide Based on ICD-10 codes from Death register | People with a main DP diagnosis of 'depressive disorders' had a greater risk of subsequent suicide and suicide attempt than those with a diagnosis of 'stress related disorders'. Substance abuse or personality disorders as a secondary DP diagnosis was statistically associated with suicide attempt in men, women and across age group, and suicide in women and younger individuals. People in receipt of a full-time DP had a higher risk of suicide attempt than those receiving a part-time DP in women and across age groups. |

(*Continued*)

**Table 3.** (Continued)

| Author (Year) & Location | Study Design & Follow-up period | Data sources | Sample size & percent female | Sample definition (exposed group) | Sample definition (Comparison group) | Primary outcome/s | Main finding |
|---|---|---|---|---|---|---|---|
| Schneider et al (2011) Germany | Retrospective Case Control Study Not reported | Key informant interviews | N=559 total Female 40.6% | Adults (age range not stated) in receipt of a disability pension (period not stated) | Community controls matched by age, gender and residential location | Suicide Based on ICD-10 codes from Death register | This study observed a significantly increased risk of suicide in the DP group compared to the healthy community control group. |

Note: N = Number; MSD = Musculoskeletal disorders. ICD-10 = International Classification of Diseases version 10.

countries, including seven from Sweden, two from Finland and a single study from each of the United States, Australia, Germany and England. Seven studies did not specify a specific condition and included samples of people with work disability from diverse causes while four studies included samples of people with mental health conditions.

Overall, these studies demonstrate an elevated risk of suicide and self-harm among people with work disability receiving a DP. Four prospective cohort studies showed a higher rate of suicide or self-harm during follow-up among people receiving a DP during the study inception period, compared to those not receiving a DP [53–56]. All of these studies observed that people with work disability due to mental health conditions are at greater risk of suicide, while one also reported elevated risk of self-harm in those with mental health conditions [55], and three reported elevated risk of suicide among people with DP due to other conditions including musculoskeletal disorders [53], and suicide attempt in young people with somatic disorders [54, 55]. A single prospective cohort study observed significantly decreased suicide mortality among people with schizophrenia receiving a DP compared to people with schizophrenia not receiving a DP [57], while a further prospective cohort study observed no association between receiving a DP in 2004 and suicide or self-harm among people with multiple sclerosis during a 6 year follow-up period [58].

Two further prospective cohort studies examined predictors of self-harm and suicide among Swedes receiving DP with common mental health conditions [59, 60]. These studies observe a higher risk of suicide and suicide attempt among those with prior suicide attempt, living alone and who had been prescribed antidepressants or anxiolytic medicines. Additional risk factors for suicide included female sex, younger age and history of inpatient care, while males had higher risk of suicide mortality. People whose primary DP diagnoses was depressive disorders had the greatest risk of suicide and suicide attempt. Two cross sectional studies observed that people receiving DP were more likely to report a history of suicidal thoughts or intentional self-harm. These included one study of Australian DP recipients [61] and a second study of Swedish DP recipients with Usher syndrome [62].

Two retrospective case-control studies observed greater odds of suicide in low income people receiving DP compared to those not receiving DP [63], and an increased risk of suicide among German adults receiving a DP compared to a healthy community control group [64]. Finally, a single ecological study conducted in England observed an increase in rates of suicide in those local areas where a greater proportion of the population had their eligibility for DP benefits re-assessed via a work capacity assessment [65].

Some notable gaps in this literature include few studies outside Nordic countries, few studies in specific occupational settings or groups, relatively fewer studies examining outcomes of suicidal ideation or self-harm; and a lack of qualitative studies.

## Studies of mixed samples

Five eligible studies reported relevant outcomes in mixed samples of people with work disability who had received either a SA payment or a DP. The studies are summarised in Table 4. Four reported suicide outcomes and three intentional self-harm. Four were from Sweden and the fifth from Denmark. One was in a sample of people with work disability due to mental health conditions, and the remaining four included samples of people with work disability arising from diverse conditions. These studies are consistent with those of the SA and DP studies described above, and provide further evidence of an association between work disability resulting in receipt of income support payments and later suicide and self-harm.

One population based prospective cohort study observed elevated risk of intentional self-harm and suicide in Swedish adults with and without mental health conditions receiving SA or DP benefits in 2005, compared to adults not receiving benefits [66]. This study also observed a positive association between outcomes and longer durations of work disability.

**Table 4. Summary of mixed sickness absence / disability pension studies.**

| Author (Year) & Location | Study Design & Follow-up period | Data sources | Sample size & Percent female | Study (Exposure) Group | Comparison Group | Primary outcome/s | Main finding |
|---|---|---|---|---|---|---|---|
| Bjorkenstam et al (2020) Sweden | Prospective Cohort Study 8 years | Social security; Death register; Hospital records; Social / labour market statistics | N = 4,195,058 Female 46.3% | Individuals (16 to 64 years) receiving either a sickness absence benefit or disability pension during 2005. | Individuals(16 to 64 years) NOT receiving a sickness absence benefit or a disability pension during 2005. | Suicide Based on ICD-10 codes in Death Register Intentional self-harm Based on ICD-10 codes from Hospital Records | Both sickness absence and disability pension were associated with elevated risk of intentional self-harm and suicide in individuals with and without mental health conditions<br><br>In Swedish-born, the risk of intentional self-harm and suicide increased as the duration of sickness absence days increased, in people with and without mental health conditions. |
| Niederkrodenthaler et al (2020) Sweden | Retrospective Cohort Study Not applicable | Social security; Death register; Hospital records; Social / labour market statistics | N = 42,684 Female 55.3% | Adults (20 to 64 years) hospitalised for intentional self-harm who received either a sickness absence benefit or disability pension in the year prior to hospitalisation. | none | Suicide Based on ICD-10 codes in Death Register Intentional self-harm Based on ICD-10 codes from Hospital Records | Long term sickness absence was associated with elevated risk of intentional self-harm. Receipt of disability pension was associated with elevated risk of both intentional self-harm and suicide death. |
| Qin et al (2000) Denmark | Retrospective Case control Study Not applicable | Death register; Hospital records; Social security; Labour market statistics | N = 811 cases Female 33.3% N = 79,871 controls Female 50.1% | Individuals (16 to 78 years) who completed suicide between 1982 and 1994. | Community controls matched to cases (matching variables not specified) | Suicide Based on ICD-10 codes in Death Register | Receipt of sickness absence benefit was a significant risk factor for suicide in men, but not women. Receipt of a disability pension **was not** associated with suicide in either men or women. |

(*Continued*)

**Table 4.** (Continued)

| Author (Year) & Location | Study Design & Follow-up period | Data sources | Sample size & Percent female | Study (Exposure) Group | Comparison Group | Primary outcome/s | Main finding |
|---|---|---|---|---|---|---|---|
| Wang et al (2015) Sweden | Retrospective Cohort Study Not applicable | Social security; Death register; Hospital records; Social / labour market statistics | N = 4,209 Female 29.1% | Adults (22 to 65 years) who completed suicide between 2007 and 2010 | None | Suicide Based on ICD-10 codes in Death Register | This study observed five different trajectories of work-related functional impairment in the five-year period preceding suicide. Nearly half of people completing suicide had a few months per year of sickness absence/disability pension, while one third recorded more than 10 month per year of sickness absence/disability pension. Two smaller groups had increasing trends of sickness absence/disability pension in the months preceding suicide, while the final smallest group recorded a decreasing number of months prior to suicide. |
| Wang et al (2020) Sweden | Prospective Twin Study 2 to 8 years | Social security; Death register; Hospital records; Social / labour market statistics; Participant survey | N = 4871 twin pairs Female 70.1% of twins in exposure group Female 52.9% of twins in control group | Adult twins (18 to 65 years) with mental health conditions who received a sickness absence benefit or disability pension between 2005 and 2010 | Adult twins (18 to 65 years) without sickness absence or disability pension due to mental health conditions between 2005 and 2010 | Intentional self-harm Based on ICD-10 codes from Hospital Records | This study shows that sickness absence or disability pension due to common mental health conditions predicts subsequent suicide attempt even after adjustment for familial factors. Specific CMD SA/DP diagnoses also showed higher risks of suicide attempt. This effect was greater in men than women, but significant in both. Higher risks were observed in people with depressive and anxiety disorders than in those with stress-related disorders. |

Note: N = Number; ICD-10 = International Classification of Diseases version 10.

Niederkrodenthaler et al [67] observed elevated risk of self-harm among hospitalised adults with long-term SA, and also elevated risk of self-harm or suicide among hospitalised adults in receipt of DP. A retrospective cohort study of adults who completed suicide between 2007 and 2010 observed five different trajectories of SA/DP receipt in the five-year period preceding

suicide [68]. The two largest trajectory groups included nearly half of the sample who had received on average a few months/year of SA/DP benefits, and approximately a further third of the sample who had received more than 10 months/year of SA/DP benefits. A prospective twin study observed that twins with MHC who received either SA or DP benefits between 2005 and 2010 had greater risk of self-harm than their adult twins without SA/DP receipt [69]. Finally, a Danish retrospective case-control study observed that receipt of SA benefits was a significant risk factor for suicide in men, but not women, but that receipt of a disability pension was not associated with suicide in either men or women [70].

## Discussion

The risk of suicide and intentional self-harm is elevated in people experiencing work disability. This relationship is observed among people receiving work disability income support from workers' compensation, disability pension and sickness absence systems; across multiple nations; and in people who are work disabled due to mental health conditions, musculoskeletal disorders, traumatic injury and other medical conditions including neurological disorders. Of the 44 quantitative studies included in this review, 41 showed a statistically significant positive relationship between work disability and suicide or self-harm. Only one study showed a negative (protective) relationship, and two showed no effect. While the relationship between work disability and suicide or self-harm is robust, in that it occurs in multiple settings and cohorts, we also observe that a range of health, social, demographic and administrative factors are associated with elevated risk of suicidal behaviour among people with work disability. These include longer durations of work disability, work disability arising from mental health conditions, younger age and female sex, a history of poor health, and living alone. Three qualitative studies in workers' compensation systems and an ecological study of disability pension eligibility re-assessment suggest that burdensome administrative processes in income support systems may also contribute to risk of suicidal behaviour in work disabled people.

Overall, our findings suggest an opportunity for reducing suicide and self-harm among working age people, via preventive interventions focused on those experiencing work disability. In developed nations with established work disability support systems such as workers' compensation and disability pension schemes, there are multiple opportunities for identification of those at risk, and delivery of preventive interventions along the pathway between work disability and suicide. These include at the onset of work absence, upon application for income support, during delivery of treatment and workplace rehabilitation programs, and at periods of review such as during the assessment of entitlement to ongoing benefits which are a feature of most work disability support systems in developed nations [11, 65]. At each of these points workers engage with formal systems operated by government, employers or government-appointed organisations involved in case management and healthcare delivery. These represent opportunities for identification of those most at risk, and delivery of services and supports to reduce risk [71]. For instance, to detect and monitor individuals at increased risk of suicide, organisations providing case management services could establish follow-up protocols for those whose work disability is due to mental health conditions. Our findings also indicate the need for more careful consideration of psychological supports and risk mitigation strategies during the process of worker rehabilitation, when there may be opportunities to address modifiable risk factors. These approaches could be adapted, for example, from existing screening and early intervention programs that seek to identify people with risk factors for long periods of work disability [72, 73].

Many of the studies included in this scoping review had features of rigorous observational study design. Multiple studies included using population-based data capture, linkage of

multiple data sources across health care and work disability systems, incorporated long follow-up periods beyond the period of work disability, utilised healthy community or other comparison groups, or adjusted statistical estimates for multiple potential confounding factors including health status prior to work disability. Consistent with scoping review methods we did not critically appraise included studies. However we noted some methodological limitations among the included studies, for example studies reporting only descriptive statistics [30], cross-sectional studies [50, 52, 61, 62] and studies reporting small sample sizes [30, 48, 52, 62]. Studies used a range of different outcome measures, and applied variety of statistical analysis techniques. As a consequence, study outcomes were reported variously using odds ratios, hazards ratios, risk ratios, standardised mortality ratios, trajectory models and descriptive statistics. While this makes it challenging to compare the strength of relationship between work disability and suicide/self-harm outcomes between studies, some broad trends were observed.

People with work disability due to common mental health conditions were observed in multiple studies to be at higher risk of later suicidal behaviour than people with work disability due to other conditions. For example, a population based register study by Wang et al [39] reports highest hazards of both suicide and self-harm in both male and female participants with MHCs than participants with other conditions. This pattern is repeated in multiple studies in sickness absence [37, 38, 50] and disability pension systems [53–56], and for self-harm outcomes [44, 55]. These findings are broadly consistent with the extensive literature demonstrating elevation of suicide risk in people with mental health conditions, and suggest that people with work-disability due to mental illness should be screened for suicide risk. We also observe elevated risk of suicide and self-harm among people with work disability due to other physical conditions, including those with musculoskeletal disorders, injury and neurological conditions. Notably, our review identified studies reporting elevated risks in people with musculoskeletal disorders in sickness absence systems [39], for women receiving the disability pension [53], in people with different MSD diagnoses [36], and for self-harm outcomes [39]. It is possible that comorbidity between mental health conditions and MSDs helps to explain these findings. These studies also demonstrate the importance of identifying and monitoring groups with common physical health conditions such as MSD to identify individuals who may benefit from intervention.

Multiple studies also report that people with longer durations of work disability were at significantly elevated risk. For example, Lundin et al [46] observed that the odds of suicide over a five year follow-up period in people sickness absent for more than 62 days during 1992–93 was 8.68 for women, and 4.41 for men, relative to people with 0 to 15 days sickness absence during the inception period. Similar trends were observed in healthcare workers [50], in males by Tang et al [43], in Finnish local government workers [49], in disability pensioners [66], and for self-harm outcomes [45]. Several authors propose that the duration of work disability does not only reflect the chronicity or severity of the health condition underlying work disability, but may also reflect other factors that can increase suicide risk, such as poor quality medical treatment, delayed help-seeking, presence of adverse health behaviours such as substance use and social isolation [45]. Importantly, our findings also suggest that work disability prevention interventions that reduce duration of disability, and thus both promote engagement in work and potentially limit exposure to these other adverse factors, may also contribute to reduced risk of suicide and self-harm. The common features of effective work disability prevention interventions are well-documented, with multiple studies reporting effective interventions in workers with mental health conditions and musculoskeletal disorders [74]. Longer-term follow-up of participants enrolled in work disability prevention trials appears warranted.

Our evidence synthesis grouped studies by the nature of the work disability system via which people were accessing income support. We chose this approach so that study findings

would be relevant to those involved in administering, and providing care and support, within those systems. This approach also demonstrates that there is a stronger evidence base in some systems and nations than others. We included the largest number of studies in sickness absence and then disability pension systems, with most of these studies located in European nations. In contrast there were fewer studies included from workers' compensation systems, and also from North American and Asian nations where such systems are more commonly a feature of government work disability support. The evidence base is larger when suicide is the measured outcome, than for self-harm as an outcome, where there are fewer studies. We identified even fewer studies reporting suicidal ideation in work disabled samples. There is value in filling these evidence gaps in order to better understand the relationship between work disability and suicide or self-harm within work disability systems and within nations. Further studies on suicidal ideation would provide critical information to explore the pathway from work disability through suicidal ideation to completed suicide, which may inform preventive interventions.

This scoping review provides an overview of the evidence relating to work disability, suicide and self-harm. We sought to draw together a diverse evidence base from across different systems of work disability support, to examine this relationship. As per accepted scoping review methods, we included a wide range of study designs, utilised systematic and transparent processes for searching and mapping retrieved studies. Our study is subject to the known limitations of scoping review methods, including that we did not appraise the quality of included studies, and that the broad scope of the research questions made it difficult to draw definitive conclusions regarding particular populations and subgroups. The studies included did not examine intermediate factors between work disability and mortality, which may include things such as opioid prescriptions, or the quality and nature of treatment, presenting an opportunity for further studies. Specifically, we did not identify any studies of suicide prevention in work disabled samples. This is a substantial gap in the evidence base given the findings of our review. A further limitation of the literature is that it was not often possible to determine if there was suicidal intent in those studies that reported self-harm outcomes, and thus our operational definition was inclusive to ensure broad capture of self-harm studies.

## Conclusions

The risk of suicide and intentional self-harm is elevated in people experiencing work disability. Within work disabled samples, several factors may place individuals at greater risk of later suicidal behaviour, longer durations of work disability, work disability arising from mental health conditions, younger age and female sex, a history of poor health, and living alone. The relationship between work disability and suicidal behaviour is observed consistently across nations, across work disability support systems and in people with a range of underlying health conditions. There is also some evidence that burdensome administrative processes in income support systems may also contribute to risk of suicidal behaviour in work disabled people. While further observational research is required to further understanding and fill gaps in the evidence base, this review suggests the need for governments, employers and those involved in the delivery of care and support to people with work disability to focus on identification and monitoring of those at greatest risk of suicidal behaviour, as well as suicide prevention in the highest risk.

## Supporting information

**S1 Text. Review protocol published on the Open Science Foundation website.**
(PDF)

**S2 Text. PRISMA-ScR checklist.**
(PDF)

**S3 Text. Search strategy and data extraction fields.**
(DOCX)

## Acknowledgments

The authors would like to acknowledge the contribution of Ms Karki Kajal to article searching and screening.

## Author Contributions

**Conceptualization:** Alex Collie, Shannon Elise Gray.

**Data curation:** Alex Collie.

**Formal analysis:** Alex Collie, Shannon Elise Gray.

**Funding acquisition:** Alex Collie.

**Methodology:** Alex Collie, Shannon Elise Gray.

**Project administration:** Alex Collie.

**Supervision:** Alex Collie, Shannon Elise Gray.

**Writing – original draft:** Alex Collie.

**Writing – review & editing:** Alex Collie, Shannon Elise Gray.

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
