## [Decision Letter · Decision Letter 0]

12 May 2022

PGPH-D-21-01167

The relationship between work disability and subsequent suicide or self-harm: A systematic scoping review.

Dear Dr. Collie,

Thank you for submitting your manuscript to PLOS Global Public Health. After careful consideration, we feel that it has merit but does not fully meet PLOS Global Public Health’s publication criteria as it currently stands. Therefore, we invite you to submit a revised version of the manuscript that addresses the points raised during the review process.

EDITOR: 

I congratulate the authors for this piece of work. However, after going through reviewers' comments and also reading the manuscripts i concur with both reviewers that a systematic review following PRISMA guideline is warranted. 

We look forward to receiving your revised manuscript.

Kind regards,

Reginald Quansah, Ph.D.

Academic Editor

Journal Requirements:

1. We ask that a manuscript source file is provided at Revision. Please upload your manuscript file as a .doc, .docx, .rtf or .tex.

2. Please provide an Author Summary. This should appear in your manuscript between the Abstract (if applicable) and the Introduction, and should be 150–200 words long. The aim should be to make your findings accessible to a wide audience that includes both scientists and non-scientists. Sample summaries can be found on our website under Submission Guidelines: 

https://journals.plos.org/globalpublichealth/s/submission-guidelines#loc-parts-of-a-submission

3. Please provide separate figure files in .tif or .eps format and removed from the manuscript file.

4. We notice that your supplementary [figures/tables] are included in the manuscript file. Please remove them and upload them with the file type 'Supporting Information'. Please ensure that each Supporting Information file has a legend listed in the manuscript after the references list.

Additional Editor Comments (if provided):

Reviewers' comments:

Reviewer's Responses to Questions

**Comments to the Author**

1. Does this manuscript meet PLOS Global Public Health’s publication criteria? Is the manuscript technically sound, and do the data support the conclusions? The manuscript must describe methodologically and ethically rigorous research with conclusions that are appropriately drawn based on the data presented.

Reviewer #1: Yes

Reviewer #2: Yes

2. Has the statistical analysis been performed appropriately and rigorously?

Reviewer #1: N/A

Reviewer #2: N/A

3. Have the authors made all data underlying the findings in their manuscript fully available (please refer to the Data Availability Statement at the start of the manuscript PDF file)?

Reviewer #1: No

Reviewer #2: No

4. Is the manuscript presented in an intelligible fashion and written in standard English?

Reviewer #1: Yes

Reviewer #2: Yes

5. Review Comments to the Author

Reviewer #1: The current scoping review has compiled the literature on the association between work disability and suicidal/ self-harm outcomes, and used a narrative analysis to explore the nature and outcome of those studies.

Title: Clear and adequate

Abstract: Under Methods, please mention the databases from where the data were obtained, along with other data sources.

Introduction: Adequate

Methods:

In the manuscript, individuals of 15 years and more with working disability are included, but in the proposal used for registration, the lower age limit was mentioned as 16. Is there any reason for change this criterion? If yes, please justify the reason for the choice of the age range.

If supplementary table I (page 8, line 171) is the Appendix I, please name it and place it accordingly.

Did the authors check the overlap of data obtained through three different methods – systematic search, reference chaining, and expert recommendations? If the systematic search from major databases combined with a manual search would yield sufficient data for a systematic review, what was the purpose of requesting expert recommendations? Considering that the study focuses on a specific research question, there was a possibility of systematic review, adopting PRISMA, that would have enhanced the robustness of the study and its outcome. So, why did the authors opt for scoping review? Please describe the reason in the Methods section with justification.

Results:

In page 14, under sample definition, age range from 15 is mentioned twice as ‘adults’. The same is repeated in Table 3, from the age of 16. Do the authors consider participants aged 15 or 16 as adults? In such contexts, it is better to replace the word ‘adults’ with ‘individuals’ or ‘participants’.

Did the authors call out DP and SA? Please include the full form of the acronyms at the first time of its appearance in the manuscript.

Discussion: Adequate

Conclusion: The major finding(s) of the current review other than the association between work disability and self-harm tendency should be highlighted, especially because it is already an established fact as the current study has also pointed out.

Reviewer #2: Overview:

This is a useful scooping review which has some good implications for practice and research. However, there are some areas which I feel the authors could benefit from, which would drive home the systematic nature of the review.

Primarily these are streamlining arguments and critical discussion, reviewing their screening process to ensure missing papers are accounted for, adding additional details to the methodology, and finally adding critical appraisal to the review.

General comments:

- Variation in terms used; scooping systematic review, systematic review – need consistency

Abstract:

- Authors could dedicate some space to the methods; which databases were used?

- Add protocol OSF link to methods and PRISMA-ScR.

Introduction:

- The introduction could benefit from being more specific (e.g. what conditions are associated with work disability? What are the studies which show substantial proportion of people have mental health conditions?) and highlighting evidence to support their arguments more closely (e.g. line 113; “financial hardship is frequently cited as a risk factor for suicidal behaviour.” But there is no reference following this statement)

- Paragraph 3 which focuses on linking work disability and suicide/self-harm seems a bit unfocused. Authors have a really nice argument in paragraph 4 (lines 125 – 128), potentially if this started paragraph 3, it would more neatly set up why paragraph 3 seems draws on risk factors rather than focus suicide data.

- Paragraph 4; authors may want to include a definition of self-harm (and suicide), and rationale as to why these are used.

Methods:

- Double referencing on line 140

- Was non-suicidal self-injury included in self-harm? This is unclear with the current eligibility criteria.

- Abstract mentions workers above 15, was this a selection criterion or a finding?

- Selection of sources of evidence: add numbers of paper selection at each stage. What was the percentage of author agreement between papers?

- Why wasn’t there any critical appraisal of individual sources of evidence? This is asked for in the PRISMA-ScR and standard across reviews. Would recommend inclusion of this.

- I would expect PRISMA flow chart in the methods, as well as paragraph 1 from Results.

- PRISMA flowchart 524-457 = 67, chart shows 65 appear to have lost 2 papers.

- What was the analysis method for the narrative review?

- Is extracted data fully available? Tell us in the methods!

- The methods feels a bit thin – I’d like to see rationale as to why authors made specific decisions.

Results:

- A large section of results is descriptive of the papers, a lot of this is summarised in tables. Authors could streamline this to enhance engagement of the reader.

- Results would benefit from critical review of the data instead of simply listing individual study findings.

Discussion:

- No critical appraisal was discussed but then quality issue is mentioned in the discussion. To be systematic this should be done with all papers with clear indication of results. The appraisal tool should be available as a supplementary material so that readers can see the criteria for high-mod-low quality. This would support reproducibility of this paper as well.

- Was there any difference between self-harm and suicide related results? Any differences between behaviour and ideation.

- Overall, arguments are clear. I think again, authors are a bit descriptive of the papers results, rather than critically discussing them and how they draw together to form implications. However, there is some clear implications from this study which is great. Pinpointing the these are very worthwhile.

6. PLOS authors have the option to publish the peer review history of their article (what does this mean?). If published, this will include your full peer review and any attached files.

**Do you want your identity to be public for this peer review?** For information about this choice, including consent withdrawal, please see our Privacy Policy.

Reviewer #1: **Yes: **Dr Allen Joshua George

Reviewer #2: **Yes: **A. Jess Williams

---

## [Author Response · Author response to Decision Letter 0]

14 Aug 2022

Dear Editor, 

We have attached a detailed response to reviewers to this revision. I hope that document is acceptable. 

Sincerely

Alex Collie

---

## [Decision Letter · Decision Letter 1]

15 Nov 2022

The relationship between work disability and subsequent suicide or self-harm: A scoping review.

PGPH-D-21-01167R1

Dear Prof Alex Collie

We are pleased to inform you that your manuscript 'The relationship between work disability and subsequent suicide or self-harm: A scoping review.' has been provisionally accepted for publication in PLOS Global Public Health.

Best regards,

Nandita Saikia, PhD

Academic Editor

Reviewer Comments (if any, and for reference):

Reviewer's Responses to Questions

**Comments to the Author**

1. If the authors have adequately addressed your comments raised in a previous round of review and you feel that this manuscript is now acceptable for publication, you may indicate that here to bypass the “Comments to the Author” section, enter your conflict of interest statement in the “Confidential to Editor” section, and submit your "Accept" recommendation.

Reviewer #1: All comments have been addressed

2. Does this manuscript meet PLOS Global Public Health’s publication criteria? Is the manuscript technically sound, and do the data support the conclusions? The manuscript must describe methodologically and ethically rigorous research with conclusions that are appropriately drawn based on the data presented.

Reviewer #1: Yes

3. Has the statistical analysis been performed appropriately and rigorously?

Reviewer #1: N/A

4. Have the authors made all data underlying the findings in their manuscript fully available (please refer to the Data Availability Statement at the start of the manuscript PDF file)?

Reviewer #1: Yes

5. Is the manuscript presented in an intelligible fashion and written in standard English?

Reviewer #1: Yes

6. Review Comments to the Author

Reviewer #1: The authors have revised the manuscript that enhanced its quality and clarity.

7. PLOS authors have the option to publish the peer review history of their article (what does this mean?). If published, this will include your full peer review and any attached files.

**Do you want your identity to be public for this peer review?** For information about this choice, including consent withdrawal, please see our Privacy Policy.

Reviewer #1: **Yes: **Allen Joshua George
